# The Role and Regulation of the NKG2D/NKG2D Ligand System in Cancer

**DOI:** 10.3390/biology12081079

**Published:** 2023-08-02

**Authors:** Ge Tan, Katelyn M. Spillane, John Maher

**Affiliations:** 1CAR Mechanics Group, Guy’s Cancer Centre, School of Cancer and Pharmaceutical Sciences, King’s College London, Great Maze Pond, London SE1 9RT, UK; ge.tan@kcl.ac.uk; 2Department of Physics, King’s College, London WC2R 2LS, UK; katelyn.spillane@kcl.ac.uk; 3Department of Immunology, Eastbourne Hospital, Kings Drive, Eastbourne BN21 2UD, UK; 4Leucid Bio Ltd., Guy’s Hospital, Great Maze Pond, London SE1 9RT, UK

**Keywords:** cancer, NKG2D, NKG2D ligands, MICA/B, ULBP, Rae-1, H60, MULT1

## Abstract

**Simple Summary:**

The immune system provides surveillance measures to identify and remove damaged cell types at an early stage. One important example involves the NKG2D receptor, which is expressed on a range of white blood cells. In humans, NKG2D binds to a family of eight proteins known as NKG2D ligands. NKG2D ligands are generally absent from the surfaces of healthy cells. By contrast, they are induced by various forms of cell stress, most notably DNA damage, which is very common in cancer cells. By this means, NKG2D provides a rapid response system to detect and eradicate potentially dangerous cells. Expression of NKG2D ligands on cancer cells can be boosted or reduced using a range of drugs, providing opportunities for therapeutic intervention. However, the NKG2D/NKG2D ligand system is double-edged since it can also fuel chronic inflammation which, in turn, can increase cancer development and progression.

**Abstract:**

The family of human NKG2D ligands (NKG2DL) consists of eight stress-induced molecules. Over 80% of human cancers express these ligands on the surface of tumour cells and/or associated stromal elements. In mice, NKG2D deficiency increases susceptibility to some types of cancer, implicating this system in immune surveillance for malignancy. However, NKG2DL can also be shed, released via exosomes and trapped intracellularly, leading to immunosuppressive effects. Moreover, NKG2D can enhance chronic inflammatory processes which themselves can increase cancer risk and progression. Indeed, tumours commonly deploy a range of countermeasures that can neutralise or even corrupt this surveillance system, tipping the balance away from immune control towards tumour progression. Consequently, the prognostic impact of NKG2DL expression in human cancer is variable. In this review, we consider the underlying biology and regulation of the NKG2D/NKG2DL system and its expression and role in a range of cancer types. We also consider the opportunities for pharmacological modulation of NKG2DL expression while cautioning that such interventions need to be carefully calibrated according to the biology of the specific cancer type.

## 1. Introduction

Natural killer group 2, member D (NKG2D), encoded by *Klrk1*, is an immune-activating receptor that belongs to the family of C-type lectin-like type II transmembrane proteins. It participates in both innate and adaptive immune responses and immune surveillance and is expressed on natural killer (NK) cells, invariant natural killer T (iNKT) cells, γδ T cells and CD8^+^ T cells [1]. CD4^+^ T cells do not generally express NKG2D, but its expression is inducible under pathological conditions, including rheumatoid arthritis [2] and cancer [3]. Like many other C-type lectin receptors, NKG2D engages multiple ligands (NKG2DL), which are induced by a range of cell stress events [4]. In humans, the NKG2DL family consists of two MHC class I-related polypeptides (MIC), namely, MICA and MICB, and six members of the UL16-binding protein (ULBP) family (1–6) [5] (Figure 1). However, the latter is something of a misnomer since only ULBP1 and -2, but not ULBP3–6, bind to the human cytomegalovirus (CMV) protein UL16 [6]. ULBP1–6 are also known as RAET1I, RAET1H, RAET1N, RAET1E, RAET1G and RAET1L, respectively, since they are counterparts of the mouse NKG2DL subfamily known as retinoic acid early inducible proteins (Rae—see below). All NKG2DL are distant MHC class I-like molecules which do not associate with β2 microglobulin. NKG2DL are highly polymorphic in humans (second only to MHC molecules in this respect), an attribute that affects their expression, affinity for NKG2D and disease susceptibility [7]. MICA, MICB and ULBP4 are expressed as membrane-spanning proteins, while ULBP1, ULBP3, ULBP6 and the allelic variant MICA*008 are anchored via a glycophosphatidylinositol (GPI) motif [8]. ULBP2 and ULBP5 can be expressed in either conformation [9]. Importantly, human NKG2D can discriminate between individual ligands in a manner that is enhanced by mechanical force, leading to significant variation in both signalling and functional outcomes within immune cells [10]. In the mouse, the NKG2DL family exhibits limited homology to the human system, comprising five Rae-1 family isoforms (a–e), three H60 isoforms (a–c) and Mult-1 [11]. 

Figure 2 summarises the structure of the NKG2D adaptor complex and downstream signalling pathways. NKG2D is expressed as a homodimer with short cytoplasmic domains that lack intrinsic signalling function. This dimer is incorporated into a hexameric transduction unit in combination with two signalling dimers of DNAX-activating protein (DAP) 10, owing to salt-bridge formation between complementary charged transmembrane residues within the complex [12]. As a result, ligation of NKG2D leads to the activation of phosphatidylinositol-3 kinase (PI3K) and Grb2-Vav1 pathways downstream of DAP10 [13]. In mice, but not in humans, the short isoform NKG2D splice variant can also pair with the DAP12 adaptor molecule, which contains an immune tyrosine activation motif (ITAM) and thus can deliver an activating signal [14]. In highly differentiated senescent-like human CD8^+^ T cells, NKG2D can associate with DAP12 in a sestrin-dependent manner, enabling the delivery of NKG2D-dependent cytotoxicity [15]. The partnering of NKG2D with either DAP10 alone or both DAP10 and DAP12 is absolutely required for cell surface expression and downstream signalling by the complex. 

This review presents the double-edged role of the NKG2D/NKG2DL system in cancer, from both a preclinical and clinical perspective. Regulation of NKG2DL expression and its pharmacological modulation is also considered.

## 2. NKG2D-Mediated Immune Response

The NKG2D system operates primarily in NK, CD8^+^ αβ T cells and γδ T cells. Although NK cell activation is controlled by the relative balance of inhibitory and activating signals, ligation of the NKG2D receptor by any NKG2DL is sufficient to trigger lytic synapse formation [16,17] and degranulation [18], overriding concomitant inhibitory cues [7]. NKG2D-triggered NK cell activation is further enhanced when the LFA-1 and 2B4 receptors interact with their respective ligands present on target cells [19]. Signalling via NKG2D can also enhance NK cell-mediated antibody-dependent cell-mediated cytotoxicity (ADCC) [20], including that promoted by therapeutic monoclonal antibodies such as rituximab [21]. In advanced cancer, circulating numbers of NKG2D-expressing NK cells are commonly reduced [22,23], and the cells often have impaired cytotoxic activity [22,24]. Conversely, the presence of tumour-infiltrating NK cells is associated with improved outcomes in several cancer types, including breast cancer [25], gastric carcinoma [26] and neuroblastoma [27].

Unlike NK cells, CD8^+^ T cells cannot be activated fully by NKG2D ligation alone. Instead, NKG2D functions as a co-stimulatory molecule in these cells, facilitating enhanced cytokine release upon T cell receptor stimulation [28,29]. Although NKG2D is not normally expressed by CD4^+^ T cells, NKG2D^+^ CD4^+^ T cells can accumulate in cases of chronic inflammation or cancer [3]. In cancer, these cells demonstrate immunosuppressive properties mediated by soluble Fas ligands and other immunosuppressive cytokines [3]. 

γδ T cells constitute around 5% of all T cells. The dominant subtype found in blood expresses a Vγ9Vδ2 T cell receptor, and these cells can receive both activating (e.g., cytotoxicity-promoting) and co-stimulatory signals via NKG2D [30]. Tissue-resident γδ T cells of the δ1 subset also express NKG2D and undergo activation when exposed to NKG2DL [31]. Although MICA is one of the human NKG2DL, it is also directly recognised by the T cell receptor found on some δ1 γδ T cells [32,33]. Human γδ T cells of both δ1 and δ2 subtypes are commonly found in solid tumours, and their presence is generally associated with a more favourable prognosis [34,35].

## 3. Expression of NKG2DL in Healthy Tissues

NKG2DL are typically present at low levels under homeostatic conditions, except in gastrointestinal and glandular epithelia where they are constitutively expressed [36,37]. In this context, expression is predominantly intracellular [38,39] and may change upon exposure to gut flora [40]. ULBP5 (RAET1G) isoform 1 is also highly expressed intracellularly in the anterior pituitary gland [41]. ULBP1 has been found in B cells and monocytes [42], while MICA and ULBP3 are present in bone marrow stromal cells [43]. The activation of T cells or the cytokine-mediated stimulation of monocytes and dendritic cells may also promote NKG2DL upregulation [44].

The expression of NKG2DL is subject to multiple forms of control at the level of epigenetic regulation [45], transcription, alternative mRNA splicing, post-transcriptional regulation (e.g., by microRNAs [46]), regulation of subcellular location (e.g., cytoplasmic versus cell surface) and release of soluble forms, either by cleavage or in exosomes [7,47]. These regulatory pathways are considered in greater detail below. Tight regulation of NKG2DL expression is believed to be necessary in order to prevent autoimmunity [48]. Nonetheless, given this complexity, it is perhaps unsurprising that the expression of NKG2DL at the mRNA and protein levels does not always concur [39].

## 4. Induction and Regulation of NKG2DL Expression in Cancer

NKG2DL can be upregulated in response to a range of factors operating during malignant transformation. These are summarised in Table 1 and described in greater detail below. 

**Table 1 biology-12-01079-t001:** Stimuli that induce NKG2D ligand expression in cancer.

Stimulus	Subtype/Notes	References
DNA-damaging agents	Radiation	[49,50,51]
	Mitomycin C	[50]
	Hydroxyurea	[50]
	5-Fluorouracil	[50,52]
	Cisplatin	[50]
	Temozolomide	[51]
	Doxorubicin	[53]
	Melphalan	[53]
	Etoposide	[53]
	Gemcitabine	[54]
	Docetaxel	[54]
	Vincristine	[55]
STING pathway		[56,57]
Cell cycle inhibition	Mediated via DNA damage response	[50]
DNA polymerase inhibitor		[50]
Oxidative stress		[58,59]
Heat shock		[32,36,49] but not [50]
Transcription factors	E2F	[60]
	KLF4	[61]
	ATF4	[62]
Oncogenic pathways	BCR/ABL	[63]
	c-myc	[64,65]
	Ras	[66]
	ErbB signalling	[67,68]
Cancer-associated metabolic alterations		[69,70]
Cancer-associated inflammation	See text	

During the process of malignant transformation, DNA damage [71], activation of heat shock proteins [72] and oxidative stress [73] all occur. Remarkably, all of these factors stimulate NKG2DL expression, emphasizing the strong link between cancer and the presence of NKG2DL. In a seminal study, Gasser et al. demonstrated that the activation of the DNA damage response (DDR) triggers the production of NKG2DL [50]. The DDR is mediated by two key protein kinases—ataxia telangiectasia mutated (ATM) and ataxia telangiectasia RAD3-related (ATR)—and it is activated upon sensing double-stranded DNA breaks and stalled replication [74]. Importantly, this pathway is further amplified by radiotherapy and by multiple cytotoxic chemotherapy agents, including 5 fluorouracil, cisplatin, gemcitabine, temozolomide and vincristine [49,50,51,52,53,54,55]. The DDR also causes the activation of P53, which in turn stimulates the transcription of ULBP1 and ULBP2 but not MICA/B [75]. Another cytosolic DNA-sensing pathway, the stimulator of interferon genes (STING) pathway, was also shown to upregulate Rae-1 expression in mice [56]. Inhibition of the STING pathway decreased Rae-1 expression in lymphoma cells and reduced their sensitivity to NK-mediated lysis [57]. 

Several additional factors of potential relevance to cancer also upregulate NKG2DL expression. Early studies showed that promoter heat shock elements regulate MICA and MICB expression [36]; consequently, the abundance of these ligands is strongly enhanced in some settings by heat shock/cell stress [32]. Moreover, the ubiquitin-dependent degradation of the murine NKG2DL Mult-1 is reduced in response to heat shock or ultraviolet radiation, providing a precedent for the post-translational regulation of NKG2DL expression [76].

It is perhaps unsurprising that certain oncogenic pathways have also been implicated in the induction of NKG2DL expression. The activation of the BCR/ABL oncogenic pathway has been linked to increased NKG2DL expression in chronic myeloid leukaemia [63], while c-myc overexpression has been implicated in NKG2DL upregulation in both lymphoma [64] and AML [65]. Mutant ras can also promote the upregulation of NKG2DL in a manner that at least partially depends on PI3K [66]. However, some of these effects may not necessarily be direct but rather may require additional genetic events in these cells. Illustrating this, the expression of K-ras and c-myc or Akt and c-myc did not induce NKG2DL expression in ovarian epithelial cells [50]. Instead, NKG2DL upregulation was only observed when these cells were injected into mice and allowed to form tumours. Similarly, in Eµ-myc transgenic animals, lymphoma formation was dependent upon additional mutations, which variably influenced expression of the NKG2DL Mult-1 [77]. 

Expression of MICA/B is also upregulated by oxidative stress [58,59] in an Erk-dependent manner [58], while p38 MAPK (mitogen-activated protein kinase) can also stimulate NKG2DL expression in some circumstances [55]. The combination of oxidative stress and Akt activation has recently been implicated in the ability of an antifungal agent (ciclopirox olamine) to increase NKG2DL expression by leukaemic cells [78]. 

Uncontrolled receptor signalling constitutes another cancer-associated process that increases NKG2DL expression. Illustrating this, the co-expression of the HER2/HER3 heterodimer resulted in the enhanced expression of MICA/B in breast cancer cell lines [67]. In both cases, PI3K signalling was implicated. Moreover, heightened EGF receptor activity has also been linked to NKG2DL upregulation [68].

A further broad stimulus to NKG2DL expression is cellular senescence [79], which is considered to be an emerging hallmark of cancer [80]. Adding complexity, senescent tumour cells may also increase NKG2DL shedding, favouring immune escape [81].

A number of transcription factors have been implicated in the regulation of NKG2DL expression. In the mouse, E2F transcription factors which promote cell cycle progression can direct the transcriptional upregulation of Rae-1 family members [60]. A similar process was inferred in the human system by virtue of reduced MICA/B and ULBP2 expression in serum-starved HCT116 cells [60]. It is also notable that E2F is a direct phosphorylation target of the ATM and ATR kinases mentioned above [44]. The KLF4 transcription factor also has been linked to the expression of MICA in acute myeloid leukaemia (AML) [61]. ULBP1 transcription is triggered by the ATF4 transcription factor, which is induced in response to nutrient deprivation, the unfolded protein response and oxidative stress [62]. However, MICA/B expression may be inhibited by the unfolded protein response under some circumstances, once again demonstrating the complex and context-dependent nature of NKG2DL regulation [82]. 

Metabolic rewiring is another distinctive feature of cancer [71]. Once again, NKG2DL expression is influenced by cancer-associated metabolic factors such as altered glycosylation [69,70].

Chronic inflammation is a key underpinning factor in the progression of many human cancers [83]. A number of inflammatory cytokines have been implicated in the control of NKG2DL expression. These include TNF-α and IL-18, both of which can upregulate ULBP2 levels in leukaemic cells [84]. NKG2DL are also upregulated by Toll-like receptor stimulation [85]. On the other hand, interferon (IFN)-γ has been shown to reduce NKG2DL on some tumour cell types, acting via STAT1 [86], microRNA induction [87] and MMP9 cleavage [88]. Similar inhibitory effects have been attributed to IFN-α [89], although there are also reports of NKG2DL upregulation in response to this cytokine [88]. Interleukin (IL)-6 and its downstream mediator, STAT3, have also been implicated in the downregulation of NKG2DL on tumour cells [90,91,92]. The effects of IL-10 on NKG2DL expression are complex: it downregulates MICA and upregulates MICB expression on melanoma cells [93] and increases NKG2DL levels on macrophages [94]. Transforming growth factor (TGF)-β downregulates the expression of NKG2DL on some tumour cell types [95,96]. Once again, however, this may not be a universal effect since the induction of tumour-associated epithelial-to-mesenchymal transition (EMT) by TGF-β may either upregulate [97] or downregulate [98] NKG2DL in a context-dependent manner. 

Tumour cells can also influence NKG2DL expression on stromal cells. Illustrating this, the release of lactate dehydrogenase 5 by glioblastoma cells induces NKG2DL expression on monocytes which in turn causes NKG2D downregulation on NK cells [99]. Moreover, tumour-associated immune infiltrates and fibrovascular structures are commonly positive for NKG2DL, particularly the membrane of endothelial cells [100].

Despite the frequency with which NKG2DL are expressed in transformed cells, levels found in malignant stem cells may be reduced or absent [101,102]. In the case of AML stem cells, this reduction could be overcome using PARP (poly-ADP-ribose-polymerase 1) inhibitors [102]. Similarly, both ULBP1 and ULBP3 are repressed in glioma stem cells that contain mutations in isocitrate dehydrogenase (IDH) genes [103]. Nonetheless, other studies have confirmed that NKG2DL remain expressed on cancer stem cells in some settings (including glioma stem cells [104]) and contribute to their susceptibility to NK cell-mediated killing [105,106,107].

Finally, it should also be noted that NKG2D itself is also subject to cytokine-mediated regulation with increased expression in response to IL-2, IL-7, IL-12, IL-15 and type 1 interferons [108]. By contrast, the reduced expression of NKG2D has been linked to IL-21 and TGF-β exposure [108].

## 5. Tumour Evasion of NKG2D-Mediated Immune Surveillance

To counteract the above, cancers have evolved various mechanisms to evade NKG2D-dependent immune surveillance. Epigenetic repression of NKG2DL expression is mediated by several pathways, including histone deacetylation, enhancer of zeste homolog 2 and DNA methylation [109]. The cleavage of NKG2DL from the surfaces of tumour cells is also an important regulatory mechanism. Soluble NKG2DL are usually present at low levels in the circulation of healthy individuals. However, levels may be highly increased in cancer patients, reaching ng/mL concentrations on some occasions [110]. The release of soluble tumour-associated NKG2DL provides a potent mechanism to downregulate NKG2D on intratumoural CD8^+^ T cells and peripheral blood mononuclear cells (including NK cells) [111,112,113]. Elevated serum levels of soluble NKG2DL have been linked to worsened patient outcomes in several cancer types [114,115,116], although, in some cases, adverse prognosis is not directly linked to NKG2D downmodulation [117]. Moreover, patients who develop autoantibodies against MICA following anti-CTLA4 immunotherapy benefitted from a reduction in soluble MICA, restoration of NK and CD8^+^ T cell function and enhanced tumour lysis and dendritic cell cross-presentation [118]. 

There are two major pathways by which soluble NKG2DL are generated in cancer. First, NKG2DL undergo cleavage by ADAMs (a disintegrin and metalloproteinases) 10 and 17 and MMPs (matrix metalloproteinases), enzymes that are commonly increased in cancer [7,119,120]. NKG2DL may also be secreted within exosomes [7,121] or extracellular vesicles (EVs) that also contain pro-apoptotic molecules such as the TRAIL (tumour necrosis factor-related apoptosis-inducing ligand) and the Fas ligand [122]. The relative contribution of cleaved and vesicle-derived soluble NKG2DL remains poorly characterized [123]. Membrane-spanning NKG2DL are primarily shed following proteolytic cleavage, while GPI-anchored NKG2DL mainly undergo release via EV. Nonetheless, both transmembrane- and GPI-anchored ligands may be found in EVs in various systems [9].

Intracellular retention of NKG2DL is another potential mechanism used to evade immune surveillance. MICA may be retained within the endoplasmic reticulum in some tumour types, in a manner that could be reversed using the proteasome inhibitor bortezomib [124]. Intracellular retention of NKG2DL may also be promoted by a number of CMV proteins [125] and by CEACAM1 [126]. 

A third recently described mechanism by which tumour cells can reduce MICA and MICB expression involves neddylation [127]. This entails the addition of a ubiquitin-like protein known as neuronal precursor cell-expressed developmentally downregulated protein (NEDD) 8, leading to protein degradation. 

A further factor that can influence the outcome of the interaction between NKG2D and its ligands is trogocytosis, a process involving the acquisition of membrane and membrane proteins from other cells during cell-to-cell interactions. Both T cells [128] and NK cells [129] can trogocytose NKG2DL from other cell types, leading to varied outcomes including enhanced NK cell activation or failure of immune surveillance owing to the death of these cells via fratricide. More recently, the transfer of NKG2DL via EVs has been demonstrated in a multiple myeloma model. Once again, a double-edged outcome can be envisioned whereby cross-dressing of tumour cells may passively sensitise them to NKG2D-dependent elimination, but NKG2D downregulation and the sensitisation/fratricide of immune effector cells could thwart immune surveillance [9]. 

## 6. Role of the NKG2D/NKG2DL Axis in Animal Models of Cancer

As indicated above, the NKG2D/NKG2DL system is believed to play a critical role in the elimination of premalignant cells before they progress into clinically detectable tumours [130]. In agreement with this, when NKG2DL are expressed on a range of malignant cell types, they facilitate tumour rejection in vivo [131,132]. 

The role of NKG2D in tumour immune surveillance is strongly supported by the fact that NKG2D-deficient mice are more susceptible to spontaneous tumour development in the TRAMP (transgenic adenocarcinoma of mouse prostate) and Eµ-myc lymphoma transgenic model systems [133]. In the TRAMP model, but not in the Eµ-myc model, tumours arising in NKG2D-sufficient mice were depleted of NKG2DL when compared to those in NKG2D-deficient mice. This highlights a divergence in mechanisms by which NKG2D immune surveillance is bypassed in both models. Adding further complexity, although antibody-mediated NKG2D neutralisation promotes enhanced sarcoma formation in response to the chemical carcinogen 3-methylcholanthrene (3-MC) [134], NKG2D deficiency did not phenocopy this effect. Indeed, there was a small trend in the opposite direction in this more slowly evolving tumour type [133]. 

A subsequent study using the TRAMP model provided additional Insights into these findings. Although there is limited cross-species reactivity between the human and mouse NKG2D/NKG2DL systems, human MICB can be recognised by mouse NKG2D. Exploiting this, bitransgenic mice were derived in which human MICB or a noncleavable derivative called MICB.A2 were co-expressed in the prostate gland of TRAMP mice [135]. Remarkably, while MICB TRAMP mice displayed accelerated disease onset compared to TRAMP-only counterparts, mice in which MICB.A2 was co-expressed were largely protected from tumour formation. Investigating mechanisms that underlie these diametrically opposite effects, it emerged that soluble MICB was responsible for the depletion of NK cells and attenuated NKG2D immune surveillance in TRAMP MICB mice. In marked contrast, membrane-anchored MICB.A2 TRAMP mice had delayed disease onset owing to the potentiation of this immune surveillance pathway. These findings suggest that the balance between membrane-anchored and soluble NKG2DL is an important determinant of prognostic impact, at least in some tumour types. Consistent with this, soluble NKG2DL is a biomarker that is often (but not always) associated with poorer outcomes in human cancer [130,136].

The maintained expression of NKG2DL is associated with a number of autoimmune and chronic inflammatory disease states, including rheumatoid arthritis, inflammatory bowel disease, type 1 diabetes, demyelinating conditions and coeliac disease [137]. It is noteworthy in this respect that chronic inflammation is a pathological process that can also facilitate malignant transformation [83]. Chronic activation of NKG2D can also accelerate tumourigenesis under some circumstances. Illustrating this, the sustained transgenic expression of an NKG2DL in vivo in a mouse model led to widespread NKG2D downregulation [138], reduced NK cell cytotoxicity mediated via NKG2D [139] and alternative receptor systems [140], sustained NK cell IFN-γ production [139] and enhanced susceptibility to chemically induced squamous cell carcinoma formation [138]. In a similar vein, the onset of diethylnitrosamine-induced hepatocellular carcinoma (HCC) was delayed in NKG2D-deficient mice [141]. 

To explain the divergent “friend” or “foe” role played by NKG2D in these various model systems, it has been suggested that if tumour rejection does not proceed efficiently, NKG2D-mediated aggravation of a smouldering chronic inflammatory process may ultimately prove protumourigenic [141]. In support of this, Sheppard et al. referred to the trend towards a detrimental effect of NKG2D in the slower-onset 3-MC sarcoma model [130], as noted above. They also commented that, in the TRAMP model, the protective effect of NKG2D was only seen in early-onset aggressive tumours. By contrast, delayed-onset tumours in these mice retained NKG2DL expression and tended to occur earlier in NKG2D-sufficient compared to NKG2D-deficient mice. Furthermore, a similar protective effect of NKG2D deficiency was also observed in the *Apc*^min^ model of colorectal cancer [142]. These preclinical findings highlight the complex relationship between the NKG2D/NKG2DL axis and cancer development.

## 7. Pharmacological Regulation of NKG2DL Expression

NKG2DL can be regulated by a wide spectrum of pharmacological agents. Upregulation of one or more ligands has been attributed to azacytidine, trichostatin A, vitamin D3, bryostatin, all-trans retinoic acid (ATRA), proteasome inhibitors, arsenic trioxide, multiple chemotherapy agents (see Section 4 above), decitabine, multitargeted tyrosine kinase inhibitors, inosine pranobex, nutlin-3a and histone deacetylase (HDAC) inhibitors such as sodium valproate and trichostatin A [51,52,54,55,103,143,144,145,146,147,148,149,150,151,152]. Mechanistically, HDAC inhibition leads to increased MICA expression at least in part via the transcription factor KLF4 [61]. Clinically relevant NKG2DL upregulation on malignant cells has been demonstrated in patients with AML following treatment with ATRA or valproic acid-containing chemotherapy regimens [153]. Moreover, in patients with pancreatic cancer who received neoadjuvant gemcitabine, MICA was expressed on 85% of tumours, in contrast to 36% of cases in the placebo-treated control group [154]. Gemcitabine also has the additional ability to reduce levels of soluble ULBP2 released by pancreatic cancer cell lines [155]. 

A further approach that may be used to increase the tumour cell surface expression of NKG2DL involves the inhibition of shedding of these ligands. Illustrating this, antibodies targeted against the α3 domain of MICA hindered the shedding of this ligand in addition to MICB [156]. As a result, NK-mediated anti-tumour activity was boosted in a number of tumour model systems. Alternatively, degradation of MICA and MICB may be inhibited using pharmacological inhibitors of neddylation [127]. Furthermore, as indicated above, the expression of NKG2DL on AML stem cells could be achieved using PARP inhibition [102].

By contrast, NKG2DL may also be downregulated using pharmacological interventions. Proteasome upregulation has been linked with the downregulated expression of ULBP1 [157]. Estradiol has been reported to either suppress [158] or stimulate NKG2DL expression [159] accompanied by enhanced ADAM 17-mediated cleavage [160]. Activation of the unfolded protein response in hepatocellular carcinoma cells also reduced the expression of MICA/B in a manner that was partially alleviated using proteasome inhibition [161]. Downregulation of NKG2DL in breast cancer cell lines has also been attributed to the anaesthetic agent sevoflurane [162]. Inhibition of BRAF with vemurafenib led to reduced MICA and ULBP2 expression by melanoma cells [163]. Rapamycin has also been linked to NKG2DL downregulation in AML [164]. Finally, the commonly used uricosuric agent allopurinol inhibited the upregulated expression of NKG2DL induced by genotoxic stress in a manner that was dependent on the inhibition of xanthine oxidoreductase [165]. In keeping with this, uric acid generated as a consequence of DNA damage and purine catabolism promoted MICA/B expression [166].

## 8. Clinical Significance of the NKG2D/NKG2DL System in Human Cancer

NKG2DL are commonly found in diverse human cancers (Table 2). However, their prognostic significance varies considerably between different studies and tumour types. Moreover, NKG2DL expression is heterogeneous not just across different cancer types but also within the same tissues and organs [167]. 

Early reports indicated that elevated MICA [168] or MICA/B and RAET1G (ULBP5) [169] were linked to improved prognosis in colorectal cancer. Importantly, although these studies were undertaken by the same group, the prognostic significance of MICA (+/−B) expression was confirmed using two different antibody reagents. The authors also observed that NKG2DL expression was highest in early-stage tumours, with a progressive decrease in the late-stage disease, consistent with the immunoediting of NKG2DL expression. More recently, however, worsened outcomes were reported in patients with colorectal tumours in which either high MICA [170] or ULBP1 (mRNA) [171] were identified. In keeping with this, the high expression of *Klrk1* in the most immunogenic consensus molecular subtype 1 (CMS1) subtype of colorectal cancer has been linked to poorer survival [142]. In breast cancer, elevated MICA expression was found more commonly in high-grade poor prognosis tumours [172]. Conversely, however, de Kruijf et al. reported that in an unselected collection of 677 breast cancers, the expression of MICA/B and ULBP2 were both associated with a significantly prolonged relapse-free interval [173]. This association was even stronger when both ligands were co-expressed.

High-intensity MICA expression has been linked to reduced survival in non-small-cell lung cancer (NSCLC) [174]. Similarly, worsened prognosis of ovarian cancer has been linked to the expression of ULBP1, ULBP3 or RAET1E (ULBP4) in univariate analysis and RAET1E (ULBP4), RAET1G (ULBP5) and ULBP2 in multivariate analysis [175]. The poor prognostic significance of ULBP2 expression in ovarian cancer was confirmed in a second independent study in which MICA/B lacked prognostic significance [176]. However, soluble ULBP2 was not detectable in these patients, suggesting that ligand shedding was not responsible for this finding. Instead, the authors noted that ULBP2 overexpression correlated with poorer infiltration of CD8^+^ T cells into tumours, raising the possibility that ULBP2 hindered T cell function via a different (contact-dependent) mechanism. Expression of ULBP2, including elevated levels of the shed form, was also associated with worsened outcomes in both melanoma and B-cell chronic lymphocytic leukaemia (CLL) [177,178]. While both soluble ULBP2 and MICA were elevated in melanoma patients, only soluble ULBP2 correlated with worsened survival. Moreover, the circulating NK cells in these subjects maintained normal levels of NKG2D expression [177]. In CLL, only soluble ULBP2 was an independent prognostic factor, while surface levels of MICA on malignant cells had no prognostic significance [178]. Once again, no reduction in NKG2D expression was observed on NK cells from these subjects, although circulating NK cell numbers were reduced, which is perhaps consistent with exosome-induced apoptosis. A separate study on ovarian cancer reported that patients had reduced numbers of NKG2D-expressing CD56 bright NK cells, while soluble MICA levels were elevated [179]. In cervical cancer, improved prognosis was linked to the expression of MICA/B and ULBP1, whereas reduced survival (univariate analysis only) was seen when tumours were RAET1E (ULBP4)- or RAET1G (ULBP5)-positive [180]. Conversely, in nasopharyngeal carcinoma, the low expression of ULBP4 was linked to worsened outcomes [181]. At the transcriptional level, the expression of MICA/B and ULBP1/2 is higher in human hepatocellular carcinomas associated with early recurrence, poorer prognosis and a less differentiated state [182]. In clear cell renal cell carcinoma, MICA and ULBP3/RAET1N were both linked to poorer prognosis, while ULBP4/RAET1E was linked to improved outcomes [23]. Where analysed, there was no correlation between NKG2DL expression on primary and metastatic tumours [183].

**Table 2 biology-12-01079-t002:** NKG2DL expression in human cancer.

Tumour	Number	MICA/B	ULBP1	ULBP2	ULBP3	ULBP4	ULBP5	Ref.
Breast (all subgroups)	677	50%	90%	99%	100%	26%	90%	[173]
Breast (all subgroups)	530	97%						[172]
Breast (no TNBC)	31	91% †	74%†	78% ***†	68% †			
Breast (ductal)	5	40%	60%	80%	60%	60%	40%	[167]
Breast	16	100%						[37]
TNBC	Not provided	93% †	85% †	85% ***†	85% †			[100] †††
Colorectal	462 *	100%	>50%	>50%	>50%	>50%	>50%	[168,169]
Colorectal	25	100% †	57% †	72% ***†	92% †			[100] †††
Colorectal	42	48%						[184]
Colorectal	5	100%	100%	80%	100%	80%	80%	[167]
Colorectal	86	85% (predominantly cytoplasmic)						[170]
Colorectal	13	100% (cytoplasmic)						[37]
AML	104	70% ^¶^						[185]
AML	50	55%						[186]
AML	30	Low-level expression seen						[42]
AML	25	0%	16%	4%	16%	0%	ND	[187]
AML	66	Preferential expression on monocytic subtypes						[188]
AML	14	0% 36% 64% 36% 14%					ND	[189]
AML/CML/CLL	25	56% expressed at least one ligand						[190]
ALL	11	0%	9%	18%	0%	0%		[187]
ALL	30	67%^¶^						[185]
CML	11	82%^¶^						[185]
CLL	3	0%	0%	0%	0%	0%		[187]
CLL	60	85%^¶^						[185]
CLL	51	Elevated MICA MFI on CLL cells						[178]
AML	50	55% ^¶¶^						[186]
T-ALL	6	5/6 ^¶¶¶^						[191]
GBM ^§^	20	94/82%	93%	84%	89%			[104]
GBM	18	88.9%		23.5%	0%	0%		[192]
Paediatric brain	125	Increased ULBP4 in low-grade gliomas only						[193]
Neuroblastoma	12–22	0%/86% ^‡^ 0% 50% 0%						[194]
CCA	82	96%	100%	77% ***				[195]
CCA	5	40%	80%	60%	60%	0%	20%	[167]
Bile duct	5	20%	40%	40%	60%	40%	40%	[167]
Ovarian	82	80%		83%				[176]
Ovarian	357	88%	63%	60%	59%	68%	85%	[175]
Ovarian (HGSOC **)	79	65%	65%	71% ***	60%			[196]
Ovarian	18	72% (cytoplasmic)						[37]
Cervical	5	20%	20%	40%	100%	80%	40%	[167]
Cervical	200	57% ^§§^	42% ^§§^	49% ^§§^	56% ^§§^	32% ^§§^	43% ^§§^	[180]
Endometrial	5	20%	60%	100%	100%	80%	10%	[167]
Melanoma	40/20 ^§§§^	78/65% ^§§§§^						[183]
Melanoma (metastases)	16	75%		50%				[177]
Bladder	23	91% †	39% †	87% ***†	78% †			[100] †††
NSCLC	91	31%	48%	50% ††	22%	69%		[197]
NSCLC	10	100% (cytoplasmic)						[37]
NSCLC	40	27.5%						[198]
NSCLC	222	98.2%						[174]
Lung AdCa	5	20%	60%	20%	40%	20%	60%	[167]
Lung squamous	5	0%	20%	20%	0%	0%	0%	[167]
Lung (unknown subtype)	6	100% (cytoplasmic)						[37]
Oesophageal	5	20%	20%	20%	20%	40%	0%	[167]
Gastric	5	60%	80%	60%	80%	80%	60%	[167]
Gastric	23	57%/50% ^‡^						[199]
Gastric	98		71%					[200]
Gastric	11	100% (cytoplasmic)						[37]
Prostate	5	0%	80%	20%	20%	20%	0%	[167]
Prostate	12	92% (cytoplasmic)						[37]
Prostate	165	65% (with 85% stromal staining which increased in Gleason stage)						[201]
Renal cell	5	0%	20%	20%	0%	0%	20%	[167]
Renal (clear cell)	71	42%						[202]
Urothelial	5	20%	60%	80%	100%	80%	60%	[167]
Tongue	5	0%	0%	0%	0%	60%	0%	[167]
Larynx	5	20%	20%	0%	0%	40%	60%	[167]
Nasopharyngeal	111	ULBP4 only measured and was reduced in tumour versus normal tissue						[181]
Thyroid papillary	5	60%	80%	60%	80%	80%	10%	[167]
Thyroid follicular	5	33%	100%	100%	67%	0%	10%	[167]
Skin	5	20%	0%	0%	0%	40%	20%	[167]
Thymoma	36	Widespread expression of all ligands; % not provided						[203]
HCC	5	40%	100%	60%	40%	60%	10%	[167]
HCC	10	60% (RT-PCR)						[204]
HCC	96	78% (not detected in surrounding noncancer tissue) ^¶¶¶¶^						[161]
HCC	54	†††† 46% 0% 0% 0%						[157]
HCC	6	50% (cytoplasmic)						[37]
HCC	143	100% MICA-positive but levels lower than in adjacent noncancer tissue						[205]
Panc. AdCa	25	63% (85% if patients had received neoadjuvant gemcitabine)						[154]
Panc. AdCa	103	89.3% (lower expression if poorly differentiated)****						[206]
Panc. AdCa	9	88% (cytoplasmic)						[37]
Panc. AdCa	22	77% (more pronounced in poorly differentiated tumours)						[207]
Panc. AdCa	5	0%	0%	0%	20%	0%	20%	[167]
Panc. AdCa	22	100% †	80% †	87% ***†	47% †			[100] †††

Abbreviations: ALL—acute lymphoblastic leukaemia; AML—acute myeloid leukaemia; CCA—cholangiocarcinoma; CML—chronic myeloid leukaemia; CLL—chronic lymphocytic leukaemia; GBM—glioblastoma; HCC—hepatocellular carcinoma; HGSOC—high-grade serous ovarian cancer; MFI—mean fluorescence intensity; NSCLC—non-small-cell lung cancer; Panc. AdCa.—pancreatic adenocarcinoma; T-ALL—T cell acute lymphoblastic leukaemia; TNBC—triple negative breast cancer. * Absolute percentage positivity not reported. These figures are for tumours with high-level expression of the indicated ligand. ** Note that this study reported the absence of ligand expression in normal control tissue, including fallopian tube epithelium and stromal cells. *** Co-staining of ULBP2/5/6. **** Positive staining noted in stroma in stage IV tumours. ^¶^ At least one NKG2DL family member present. ^¶¶^ MICA/B only analysed. ^¶¶¶^ Combined assessment of all NKG2DL. ^¶¶¶¶^ Expression intensity was reduced in more advanced tumours. ^§^ Percentages refer to GBM stem cells only. ^§§§^ High-level expression only (as all tumours were classified as high or low). ^§§§§^ Primary/metastatic. ^†^ Figures exclude weak staining. ^††^ Combined ULBP2, -5 and -6. ^†††^ These data were extracted from a poster presentation made by the authors and previously available on the Celyad Oncology website. ^††††^ MICA was mainly detected in vascular endothelial cells of well- and moderately differentiated tumours, while ULBP1 was detected in tumour cells of well- and moderately differentiated tumours but not poorly differentiated tumours. ^‡^ MICA/MICB percentages.

The contradictory findings reported in many human clinical studies suggest that the function of NKG2D/NKG2DL in tumour development and progression is highly context dependent. A favourable association may reflect the fact that cell surface NKG2DL flag tumours for the attention of NKG2D-expressing immune cells such as NK cells, CD8^+^ T cells and γδ T cells. On the other hand, NKG2D-mediated chronic inflammation may be pro-tumourigenic, accounting for the detrimental influence of NKG2DL expression in some cancer types. In addition, overstimulation of immune cells by excessive expression of NKG2DL could promote the exhaustion of immune effector cells. Furthermore, tumours in which significant NKG2DL release occurs may ultimately have worsened outcomes due to the immunosuppressive effects of soluble NKG2D ligands.

Some studies have demonstrated how NKG2D itself can also be exploited by tumour cells as a survival mechanism. Cell surface expression of NKG2D was detected in several malignancies, including ovarian, breast, colon and prostate cancers [208,209,210]. It was shown that NKG2D signalling in tumour cells could promote the acquisition of stem cell-like attributes and facilitate tumour growth, epithelial-to-mesenchymal transition and metastasis [208,209,210]. Ligation to the NKG2D ligands on adjacent tumour cells potentially activates oncogenic pathways such as PI3K and Erk cascades, which were shown to increase cell motility and survival in tumour cells [208,209,210].

## 9. Conclusions

The NKG2DL system provides a sophisticated innate immune surveillance mechanism in which multiple layers of regulation apply to balance the early detection of stressed cells while avoiding the induction of autoimmunity. The complexity of the system is driven by the highly polymorphic nature of human NKG2DL [211], multifaceted control of cell surface NKG2DL expression, ligand-specific differences in signals and functional outputs mediated by NKG2D/NKG2DL interaction and vital differences in outcome when NKG2D encounters cell-surface versus secreted NKG2DL formats. Polymorphism of NKG2D itself is also an important factor in the risk of cancer development, with high cytotoxicity-associated haplotypes being linked to reduced cancer occurrence [212]. While evidence indicates that this system can achieve prompt removal of damaged cells, failure of swift resolution may initiate a chronic, progressive and ultimately detrimental inflammatory process. Put another way, malignant cells that are placed in the spotlight of NKG2D-mediated surveillance commonly evolve countermeasures that can neutralise or even harness this pathway to accelerate disease progression. This complex interrelationship is mirrored by the fact that there is considerable variability in the clinical significance of NKG2DL expression in human cancers, with opposing results reported for similar tumour types on some occasions. While the balance between soluble and membrane-anchored versions of these ligands may be important in influencing prognostic significance, this does not provide a complete explanation for this variation, nor does the ability of some forms of soluble NKG2DL to downregulate NKG2D. Context appears to be a key factor in understanding how NKG2DL are regulated and the consequences of this process for tumour control or progression. The complex and potentially double-edged nature of this system requires very careful consideration of the use of pharmaceuticals which can perturb NKG2DL expression.

## Figures and Tables

**Figure 1 biology-12-01079-f001:**
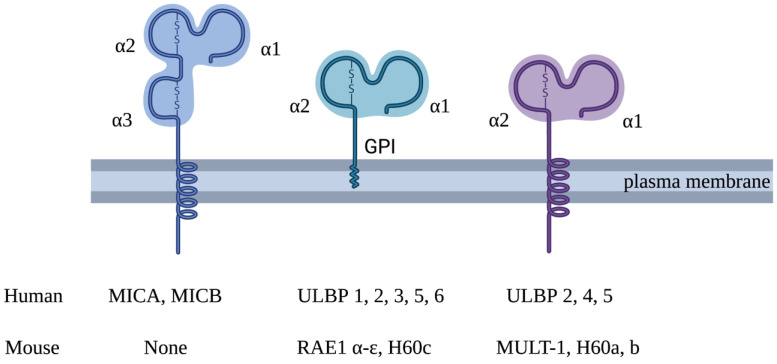
Structure of human and mouse NKG2D ligands. Human NKG2D ligands comprise MICA, MICB and ULBP1–6. MICA and MICB have three extracellular domains (α1, α2 and α3) and a transmembrane domain. Unlike the MIC family, ULBP family members lack an α3 domain and only have MHC class 1-like α1 and α2 extracellular domains. ULBP1, -3 and -6 (and the MICA*008 allelic variant) are anchored to the plasma membrane via a glycosylphosphatidylinositol (GPI) motif at the C-terminus. ULBP4 consists of a transmembrane domain and is expressed as a transmembrane protein. ULBP2 and -5 can be expressed in either conformation. NKG2D ligands in mice include Rae-1 a–e, H60 a–c and MULT-1. There are no mouse equivalents of human MICA or MICB. Rae-1 a–e and H60 c are expressed as GPI-anchored proteins, and MULT-1 and H60 a and b are expressed as transmembrane proteins. They all have MHC class 1-like α1 and α2 extracellular domains.

**Figure 2 biology-12-01079-f002:**
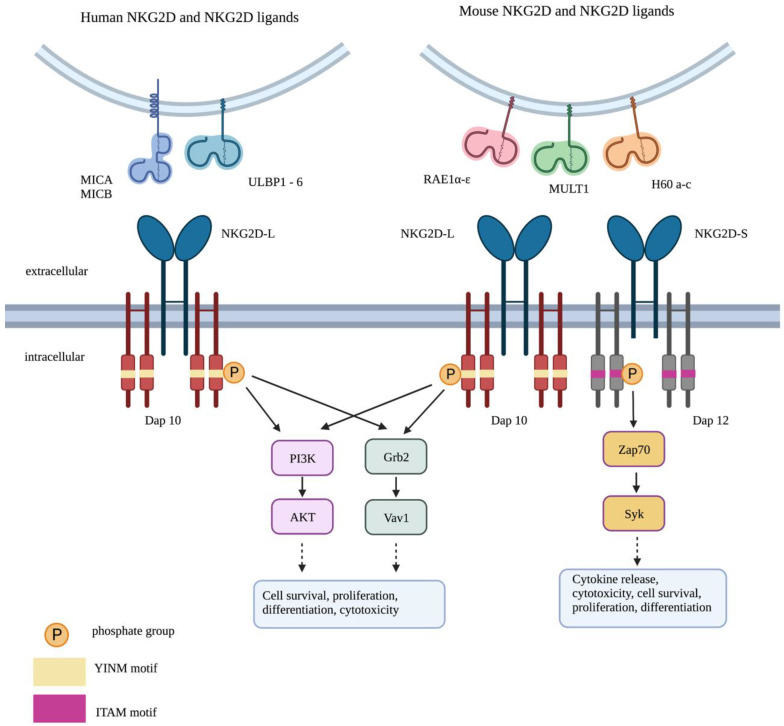
Signalling by the NKG2D system. Long (L) and short (S) isoforms of NKG2D receptors are found in the mouse, allowing interaction with DAP10 alone or both DAP10 and DAP12. By contrast, only the long isoform of NKG2D is found in human cells, allowing association with DAP10 alone. Phosphorylation of the YINM motif within DAP10 triggers the activation of two key pathways: PI3K/AKT and Grb2/Vav1. This leads to cell survival, proliferation, differentiation and cytotoxicity. Phosphorylation of the immune tyrosine activation motif (ITAM) in DAP12 activates ZAP70/Syk signalling. This results in cytokine release, cytotoxic granule secretion, cell survival, proliferation and differentiation.

## Data Availability

Not applicable.

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
