# Peer review of "The Role and Regulation of the NKG2D/NKG2D Ligand System in Cancer"

_biology, 2023, doi:10.3390/biology12081079_

Round 1
Reviewer 1 Report
Dear Authors,
Congratulations on your article "Role and Regulation of the NKG2D/NKG2D Ligand System in Cancer". The paper is relevant and interesting in terms of helping in providing opportunities for therapeutic intervention by increasing or reducing the NKG2D ligands' expression on cancer cells using a range of drugs but still has some points to be addressed or cleared.
Here you can find minor comments :
1. I noticed that there are a lot of old references. I would suggest adding more recent references.
2. In this sentence ( "Oncogenic pathways such as BCR/ABL [65] and c-myc [66,67] have been linked to increased NKG2DL expression in various tumour systems" ) when you say that oncogenic pathways have been linked to the increased expression of NKG2DL in various tumor system, it would be great if you give examples of tumors with references.
3. The idea of this review is to cite the role and regulation of the NKG2D/NKG2D Ligand System in Cancer and the most relevant is to find a new therapeutic intervention by over-expressing or inhibiting the NKG2D ligands' expression on cancer cells, that's why I suggest you to elaborate more the section 3. Pharmalogical regulation of NKG2DL expression
4. Please elaborate more on the conclusion to have a clear picture of the manuscript and the idea you want to be understood by the readers. I suggest adding an abstract scheme summarizing the general idea of the manuscript.
Thanks,
Author Response
We thank the reviewer for their helpful comments. In response to the points raised:
- We have added 6 new references from 2023 to address this balance, meaning that a total of 212 references have now been cited.
- This sentence has been rewritten to add these details e.g. "Activation of the BCR/ABL oncogenic pathway has been linked to increased NKG2DL expression in chronic myeloid leukaemia (ref), while c-myc over-expression has been implicated in NKG2DL up regulation in both lymphoma (ref) and AML (ref).
- We agree and have extended this section with discussion of five additional references.
- Both the conclusion section and abstract have been revised and extended in scope.
Reviewer 2 Report
In this manuscript, Tan et al. provide a comprehensive review on our current knowledge of the NKG2D/NKG2D Ligand system with a clear focus on cancer. Overall, the paper is very well written and optimally referenced (206 references!) with a sufficient proportion of citations of the most recent work on the topic. The authors try to render understandable the inherent high complexity of this receptor - ligand system, and succeed in this in large parts of the manuscript.
MINOR SUGGESTIONS:
1) Page 1, last line: "ULBP3" and not "ULB3"
2) Use symbols (a and not a; g and not g) to characterize the different cytokines mentioned
3) Part 4 is somehow a confusing enumeration of details and might be edited for better clarity. A summary table could be helpful here
Author Response
We thanks the reviewer for their helpful comments which have been addressed in the following manner:
- This error has been corrected.
- Greek symbols are used throughout the submitted word manuscript for cytokines such as interferon alpha, interferon gamma and transforming growth factor beta.
- Part 4 has been re-structured to enhance clarity and a summary table has been added.
Round 2
Reviewer 1 Report
Dear authors,
Thank you for answering the comments I addressed and congrats you for this relevant manuscript. The manuscript can be accepted in its present form.
Thanks,
Reviewer 2 Report
The authors satisfactorily addressed the minor comments of this reviewer. No further modifications are needed in my opinion.